# Parallel Microdispensing Method of High-Viscous Liquid Based on Electrostatic Force

**DOI:** 10.3390/mi13040545

**Published:** 2022-03-30

**Authors:** Zheng Xu, Shaochun Qin, Yu Yu, Xiaodong Wang, Junshan Liu, Wenxin He

**Affiliations:** Key Laboratory for Micro/Nano Technology and System of Liaoning Province, Dalian University of Technology, Dalian 116024, China; qinshaochun@mail.dlut.edu.cn (S.Q.); yuyu980323@mail.dlut.edu.cn (Y.Y.); xdwang@dlut.edu.cn (X.W.); liujs@dlut.edu.cn (J.L.); hewenxin@mail.dlut.edu.cn (W.H.)

**Keywords:** printing head, microdispensing, parallel transfer printing

## Abstract

Parallel microdispensing of high-viscous liquid is a fundamental task in many industrial processes. Herein, a smart printing head is developed, including the probe array, the electric control module, the contact force measurement module, and the extra force balance module. The parallel dispensing of high-viscous liquid in nL level is achieved. The interacting effect between probes on the loading process is analyzed too. According to the result, the interacting effect between probes has a strong influence on the loading process. Therefore, the strategy of serial electrical loading and parallel transfer printing is utilized. Finally, the dependency of transfer printing volume on probe size, etc., is experimentally investigated. The volume of the loaded droplet can be controlled by the lifting velocity of the probe array, and the volume of the transferred droplet can be adjusted by the size of the probe instead of the contact force. The advantage of the proposed method is to realize the highly repeatable parallel dispensing of high-viscous liquid with a relatively simple device.

## 1. Introduction

Microdispensing of high-viscous liquid is a fundamental task in various industrial productions [1,2,3]. Taking the fabrication of microdevices as an example, many droplets of high-viscous adhesive are required for the connection and encapsulation process [4,5]. To improve efficiency, the development of parallel microdispensing is imperative, which can permit many droplets to be efficiently transferred onto the target. The main challenges for parallel microdispensing are resolution and reliability. Currently, the transfer printing technique is fairly convenient for parallel microdispensing [6,7,8]. Its core can be described as: with a given mass of liquid between two surfaces, called the liquid bridge, some liquid is transferred from one probe or stamp onto another surface as the two surfaces are separated by the stretching action [9,10]. The transfer printing approach overcomes the high flow resistance problem in the injection approach, thereby providing a promising solution to parallel microdispensing. Transfer printing can be coarsely divided into droplet loading onto a stamp or probe and the droplet transferring onto the substrate. To improve resolution and reliability, there has been considerable effort involving the development of a stamp or probe and control strategy of the process [11,12,13].

In transfer printing, the elastomeric PDMS stamp is widely used to transfer ink onto a substrate. Until now, most stamps in the literature have been made of polydimethylsiloxane (PDMS) (Sylgard 184, Dow Corning) that molds with high fidelity to a patterned template [1,14,15]. Moreover, as PDMS is an elastomeric material, the stamp deforms to allow the raised features to conform to the substrate [16,17]. Lastly, PDMS does not react with many chemicals. Besides elastomeric stamps, some hard pins made of metal materials are also used for transfer printing [1].

In terms of the control strategy, the stretching velocity of the stamp and the contact force between the stamp and substrate have been widely investigated since they can influence the transferring ratio in most phenomena of liquid-bridge breakup. For example, Hong et al. used an atomic force microscope to obtain the contact pressure. With different contact pressure as the threshold value for stretching action, the dot sizes can be adjusted [18]. Kim et al. found that as the contact force decreased from 1.0 to 0.2 mN, the area of the transferred ink droplet as a percentage of the bottom area of the elastic microtip cone dramatically decreased from 80% to 0.7% [19]. Liao et al. found that the diameter of transferred ink drops decreases from 1.99 to 0.56 μm with the decrease in contact force from 15.9 to 0.7 mN [20,21]. Khandavalli et al. studied the influence of the stretching velocity on the ink transferring ratio. Their results show that the ratio is less than 20% at low stretching velocity (0.5 mm/s), whereas the ratio achieves ~70% as the velocity increases to 10 mm/s. However, the ratio of ink transfer is nonlinear to the stretching velocity [22].

We previously proposed the electrostatic-assisted transfer printing approach described in Section 2.1 [11,23]. Compared with others [1,2,3], it is more suitable for high-viscous liquids. Herein, we further develop the parallel dispensing method based on this principle:A smart printing head is developed for parallel dispensing.The interacting effect between the probes in the loading process is analyzed.Correspondingly, the overall strategy, emphasizing serial electrical loading and parallel transfer printing, is built up.The dependency of transferred volume on probe size, etc., is investigated.

## 2. Development of Parallel Dispensing Device and Process

### 2.1. Electrostatic-Assisted Transfer Printing Principle

To promote understanding, the principle of electrostatic-assisted transfer printing is briefly described as shown in Figure 1.

Before the loading stage, the viscous liquid film is deposited onto a gold-plated silicon wafer. First, the probe array is kept at some distance from the liquid film as the donor, and then an electric potential is applied between the liquid film and the probe array. Thus, driven by the electric force, some liquid is stretched up to contact the surface of the probe array. Once all liquid bridges are formed, the probe array is lifted to break up these liquid bridges. Therefore, these droplets are loaded onto the probe array.

In the transferring stage, the probe array with these loaded droplets moves above the substrate and then slowly declines to contact the substrate. Once the contact force reaches a threshold, the probe array is lifted to break up these liquid bridges. Consequently, these droplets are partially transferred onto the substrate.

### 2.2. Parallel Dispensing Device

A smart printing head is developed for parallel dispensing as shown in Figure 2, which is mainly composed of the probe array, the electric control module, the contact force measurement module, and the extra force balance module.

Instead of an elastomeric PDMS stamp, we chose a circuit board and some metal probes to form the probe array. One reason is electro-conductibility. For electrostatic loading, the probe must be conductive. However, the PDMS metallization technique is still immature and the metal film on PDMS is easy to wrinkle. Moreover, flexibility is considered too. Various patterns can be economically obtained by replacing the circuit board and probes. Here, these metal probes were purchased from the Misumi company (Tokyo, Japan).

The electric control module provides the electrostatic force for loading liquid onto the probe array. It is composed of a high-voltage power supply, an ammeter (resolution: 0.1 nA), etc. In the electric circuit, the gold film on the silicon wafer is the negative electrode, and the probe array is the positive electrode.

The contact force measurement module is utilized to obtain the contact force between the probe array and the substrate, which is composed of a force sensor (resolution: 1 mN, range: 0~2 N), a transition plate, and a connecting rod.

To keep the force sensor in the linear range, the extra vertical stretching effect from the gravity force of the probe array, etc., onto the sensor has to be eliminated in advance. Here, we developed the extra force balance module that is composed of a spring, a guide rod, a sliding bearing, and a nut as shown in Figure 2. The balancing task is performed after the installation of the probe array as follows: by rotating the nut at the bottom, the shaft of the sliding bearing is lifted along the guide rod to compress the spring. As a result, the elastic force $F_{b}$ on the fulcrum will gradually increase until the sensor reading achieves zero, meaning the elastic force is equal to the extra force $F_{a}$.

The developed smart printing head is installed on the 3-Axis robot (YAMAHA, SXYx-S-M1, Resolution: 1 μm). The overall platform is shown in Figure 3.

Furthermore, due to the existence of elastic and frictional factors, the force sensor needs to be recalibrated with a high accuracy balance as shown in Figure 4a. When the tips of the probes begin touching the balance, the probes slowly lower and the force from the sensor and the balance are simultaneously recorded. As the printing head descends, the force gradually increases. When there are two and four 0.4 mm probes on the board, the ratio of the balance to the force sensor is 1.85. The results as shown in Figure 4b illustrate that the value of the sensor and balance increases linearly as the robot descends.

## 3. Control Method of The Loading Process

The droplet loading process is simulated to find the controlling strategy to ensure that each probe can reliably load one droplet. The epoxy-based liquid is used for the simulation, and the parameters are shown in Table 1. The simulation is performed with COMSOL software, in which the phase field module and the electrostatics module are used.

It is difficult to flatten the high-viscous liquid by spinning. Thus, it is necessary to investigate the influence of thickness difference of the liquid film on the loading process. Figure 5a shows the simulation set, in which *h*_1_ and *h*_2_ are the shortest distances from the left probe and the right probe to the liquid surface. *U*_d_ is the driving voltage. *U*_surf-1_ and *U*_surf-2_ are the electric potentials of the liquid film surface at the bottom of the two probes.

Figure 5b presents the influence of _Δ_*h* with *U*_d_ = 400 V. When _Δ_*h* is zero, the liquid can rise until it contacts both probes. When _Δ_*h* = 20 and 40 μm, the liquid at the bottom of the right probe rises until it contacts the probe, whereas the liquid in the left first rises and then falls. Take _Δ_*h* = 20 μm as an example: when *t* = 0 s, *h*_1_ = 160 μm and *h*_2_ = 140 μm. When *t* = 9 s, *h*_2_ decreases to 0 and *h*_1_ decreases to 113 μm. When *t* = 14.5 s, *h*_1_ increases to 160 μm again. Therefore, the liquid cannot rise to contact all probes if _Δ_*h* ≠ 0.

In addition, the effect of driving voltage on the droplet loading process was also investigated as shown in Figure 5c,d. When *U*_d_ = 400 and 450 V, the liquid at the bottom of the right probe rises until it contacts the probe, whereas the liquid in the left first rises and then falls. When *U*_d_ > 500 V, the liquid at the bottom of the two probes can rise to contact the probes. Although raising the voltage can solve the problem, too high voltage may induce air electric breakdown, joule heating, etc.

To ensure that each probe can reliably load one droplet, the serial loading method was employed. As shown in Figure 6a, the simulation is based on the status that the liquid contacts the right probe at _Δ_*h* = 60 μm, *U*_d_ = 400 V (*t* = 4 s). The result is shown in Figure 6b,c. When the driving voltages are 550, 600, 650 V, *h_1′_* decreases to 0 at *t* = 7.5, 5.5, 4.5 s, *U*_surf-1′_ increases to 550, 600, 650 V at *t* = 7.5, 6.0, 5.5 s, respectively.

## 4. Experimental Results and Discussion

### 4.1. Serial Electrical Loading

In this paper, we selected the scenario of loading epoxy liquid with two probes for experiments. The viscosity of the epoxy liquid is 15 Pa·s, the density is 1.2 ${g/\mathrm{cm}}^{3}$, and the dielectric constant is 2.27. Figure 7a–f shows the droplet loading process, which demonstrates the feasibility of the previously proposed control method.

(a) The probes on the printing head are moved onto the film of epoxy liquid, and then the electric power is connected with the two probes.

(b) As a result, the epoxy liquid below the probe is stretched up by the electrostatic force.

(c) Once the liquid contacts the probe surface to form the liquid bridge, the electrical current begins increasing and the liquid bridge continuously volumes up.

(d) As the rising flow is blocked by the sidewall of the probe, both the electrical current and the liquid bridge volume stabilize.

(e) The connection of electric power is switched to the probe, which is not connected with liquid. The process from (c) to (e) is repeated.

(f) The electric power is turned off, and the printing head is lifted to break up these liquid bridges. As a result, the droplets are loaded onto the probes.

The droplet volume is calculated by Equation (1):(1)$$V_{\mathrm{drop}}=\frac{1}{4}\overset{n}{\underset{i=1}{\sum }}{\pi x}_{i}{}^{2}h_{p}\;\left(i=1,\;2,\;3,\ldots \right)$$
where the droplet is deliberately divided into *n* cylinders. *V*_drop_ is the loaded volume. The unit height *h*_p_ is equivalent to the value of a single pixel, and *x_i_* is the number of pixels in the cylinders. More details can be found in our previous work [23].

Figure 8a shows the results of loaded volume with different probes, in which the driving voltage *U*_d_ is 400 V, the interval distance *a* is 5 mm, and the average thickness of the liquid *h* is 500 μm. The experiments were repeated 20 times, and the loaded volume is the average of the volume loaded by two probes. The simulation of the loading droplet is divided into two stages with COMSOL. The first stage is to use the electrostatic module and the phase field module to simulate the rise of the liquid into contact with the probe. The second stage is to use the moving mesh module and the phase field module to simulate the liquid bridge breakup process when the probe is raised. The free interface of the air–liquid is defined by the isoline with a 0.5 volume fraction of the viscous liquid, and the loaded volume is quantified by an integral [24]. As the diameter *d* of the probe is 0.40, 0.65, 0.80, and 1.00 mm, the loaded volume increases from 7.84 to 117.6 nL in the simulation. The average loaded volume increases from 7.23 to 94.94 nL, and the standard deviation is from 0.22 to 1.82 nL in the experiment. This shows that the loaded volume of the droplet strongly depends on the probe diameter.

Figure 8b shows the effects of driving voltage on the loaded volume. The experiments were performed with the following parameters: *d* = 0.65 mm, *a* = 5 mm, *h* = 500 μm. When the driving voltage increases from 400 to 650 V, the loaded volume is from 34.0 to 32.6 nL in the simulation and from 33.9 to 32.1 nL with a standard deviation of 0.22 to 0.93 nL in the experiment.

Figure 8c shows the effect of interval distance on the loaded volume, *d* = 0.65 mm, *U*_d_ = 400 V, *h* = 500 μm. When the interval distance increases from 2.5 to 6.0 mm, the loaded volume slightly varies in the range from 33.8 to 32.0 nL in the simulation, and the loaded volume varies in the range from 32.30 to 31.64 nL with a standard deviation of 0.73 to 0.77 nL in the experiment, both indicating that the droplet-loaded volume is less dependent on the driving voltage and interval distance.

Figure 8d,e shows the influence of lifting velocity on the loaded volume. The experiments were performed with the parameters: *U*_d_ = 400 V, *a* = 5 mm, *h* = 500 μm. As the lifting velocity increases from 120 to 1440 μm/s, the loaded volume increases from 7.8 to 137.0 nL. It can be inferred that the loaded volume increases with the increase in lifting velocity. This is because inertial forces start to dominate as the lifting velocity increases.

### 4.2. Parallel Transfer Printing

The transfer printing process is shown in Figure 9a–f and is described as follows:

(a) Move the probe array onto the substrate.

(b) Slowly descend the probe array to form liquid bridges between the probe array and the substrate.

(c) Once the contact force achieves the threshold, lift the probe array to break up these liquid bridges.

(d) As a result, these droplets are transferred onto a substrate. The transferred droplets are shown in Figure 9e with a 2 × 2 probe array and Figure 9f with a 4 × 2 probe array. The diameter of the probe is 0.4 mm, and the interval distance between the probes is 5.0 mm.

Figure 10 shows the dependency of transferred droplets on the contact force with the different probes. As the diameter of the probe is 0.40, 0.65, 0.80, and 1.00 mm, the average transferred volume increases from 4.66 to 60.02 nL with the standard deviation in the range from 0.30 to 2.80 nL. The mean transfer ratios are 61.20%, 58.30%, 61.84% and 63.38%, respectively. During the transfer process, the contact force is mainly composed of surface tension and viscous force. The viscous force of high-viscous liquid is more dominant than the surface tension for the nL level liquid transfer [23].

Different from other studies [19,20,21], here, the transferred volume or transfer ratio is less dependent on the contact force. Taking *d* = 1.00 mm as an example, as the contact force increases from 1 to 5 mN, the transferred volume slightly changes from 59.77 to 60.76 nL. The reason is that the metal probes cannot be deformed under the contact force. The contact area between the liquid and the substrate does not increase; thus, the transferred volume is mainly adjusted by the size of the probe instead of by the contact force. The role of the contact force is simply to trigger the breakup motion of the liquid bridge for our method.

## 5. Conclusions

The conclusion is described as follows:A smart printing head is developed, including the probe array, the electric control module, the contact force measurement module, and the extra force balance module. The parallel dispensing of the high-viscous liquid in nL level is achieved.According to the simulation results, the interacting effect between the probes has a strong influence on the loading process. Therefore, the strategy of serial electrical loading and parallel transfer printing is utilized.The dependency of transfer printing volume on probe size, etc., is experimentally investigated. The loaded volume can be controlled by the lifting velocity of the probe array, and the transferred volume can be adjusted by the size of the probe instead of by the contact force.

The main advantage of the proposed method is to realize the highly repeatable parallel dispensing of the high-viscous liquid with a relatively simple device. In the future, we will try to improve the probe density using micromanufacturing techniques.

## Figures and Tables

**Figure 1 micromachines-13-00545-f001:**
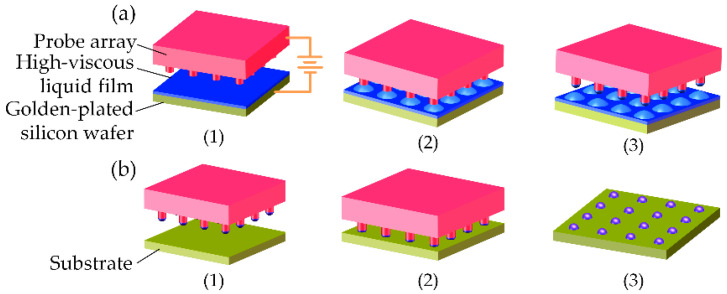
Schematic illustration of the process flow for electrostatic-assisted transfer printing. (**a**) (1)–(3) Loading Stage: forming the liquid bridges by electric force and then transferring the droplets onto the probe array; (**b**) (1)–(3) Contacting the probe array to another substrate and then slowly lifting it up to partly transfer droplets onto the substrate.

**Figure 2 micromachines-13-00545-f002:**
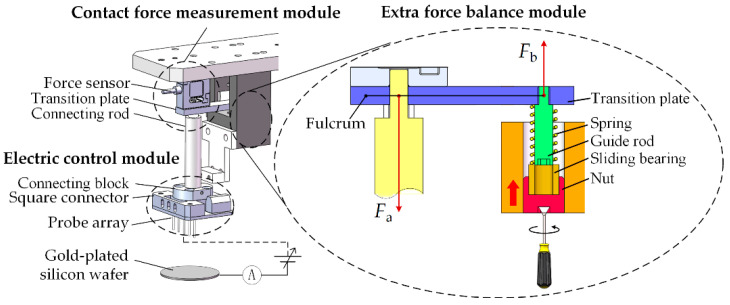
Schematic diagram of the smart printing head.

**Figure 3 micromachines-13-00545-f003:**
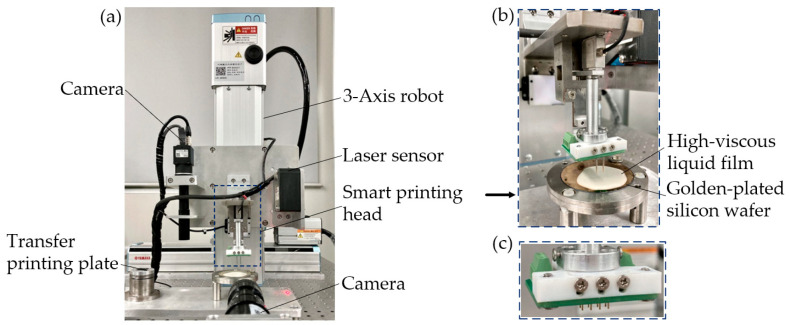
(**a**) Experimental platform; (**b**) parallel printing head; (**c**) 4 × 2 probe array on circuit board.

**Figure 4 micromachines-13-00545-f004:**
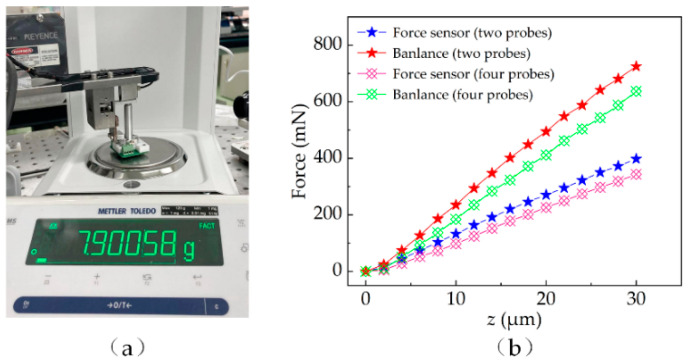
(**a**) Experimental setup of calibration; (**b**) measured results.

**Figure 5 micromachines-13-00545-f005:**
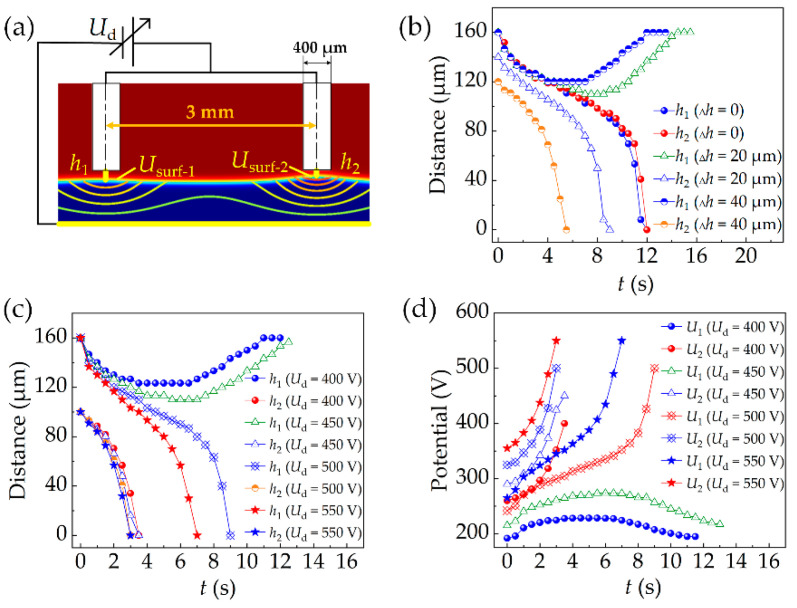
(**a**) Setting of simulation conditions. (**b**) The shortest distance between the liquid surface and the probes changes with time in different thickness differences and (**c**) different driving voltages. (**d**) Liquid surface potential changes with time in different driving voltages.

**Figure 6 micromachines-13-00545-f006:**
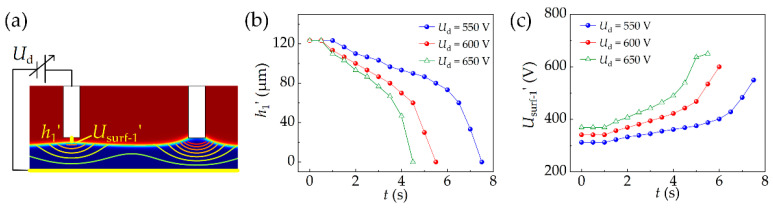
(**a**) Setting of simulation conditions. (**b**) The shortest distance between the liquid surface and the left probe changes with time in different driving voltages. (**c**) Liquid surface potential changes with time in different driving voltages.

**Figure 7 micromachines-13-00545-f007:**
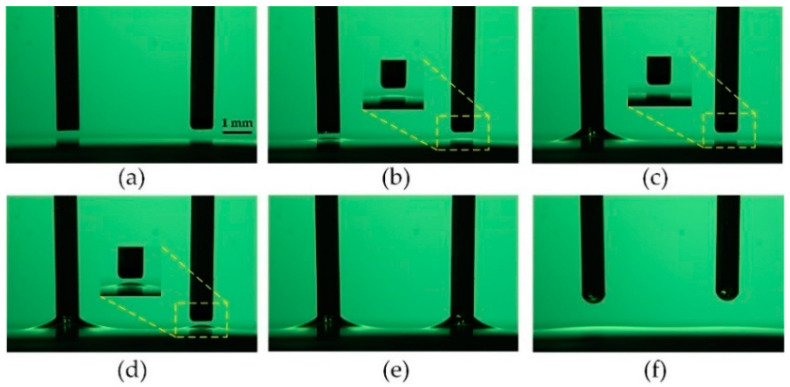
(**a**) The probes are moved onto the viscous liquid film. (**b**,**c**) The liquid below the probe is stretched up by electrostatic force until it contacts the probe. (**d,e**) The connection of electric power is switched to another probe. (**f**) The printing head is lifted to break up these liquid bridges.

**Figure 8 micromachines-13-00545-f008:**
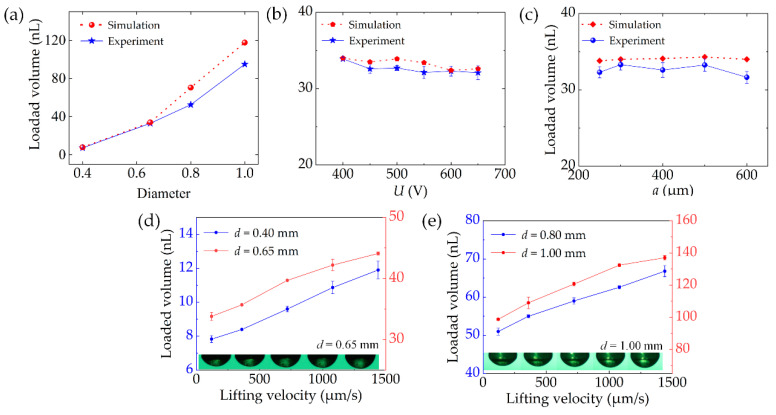
(**a**) Dependency of loaded volume on the size of the probe. (**b,c**) Results of the loaded volume at different driving voltages and interval distances. (**d,e**) Influence of lifting velocity on loaded volume.

**Figure 9 micromachines-13-00545-f009:**
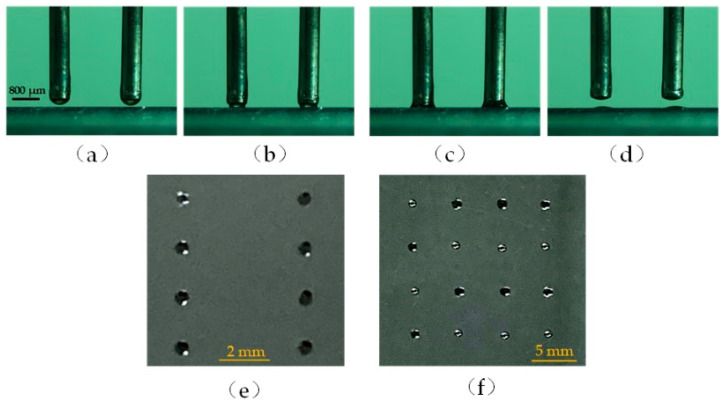
(**a**–**d**) Parallel transfer printing process. (**e**,**f**) Transferred droplets onto the substrate.

**Figure 10 micromachines-13-00545-f010:**
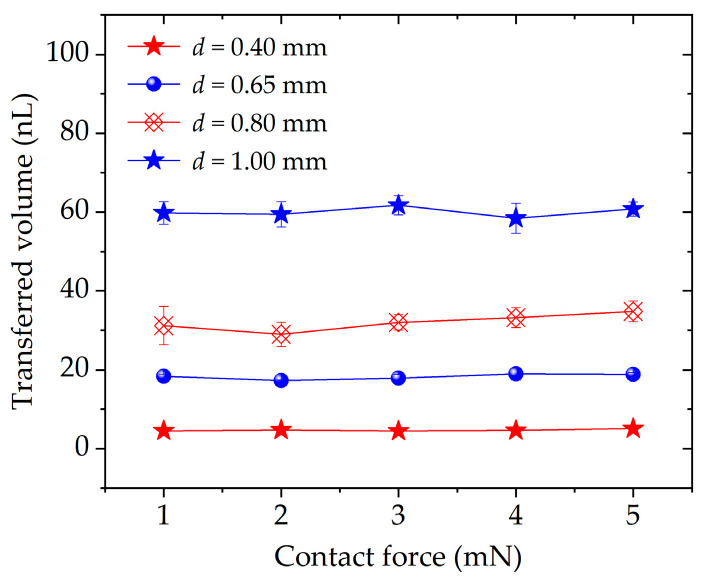
Transferred volume with different probe array.

**Table 1 micromachines-13-00545-t001:** Physical parameters for simulation.

Probe Diameter (μm)	Interval Distance between Two Probes (mm)	Driving Voltage (V)	Density (g/cm^3^)	Permittivity	Viscosity (Pa·s)
400	3.0	400~650	1.2	2.27	15

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
