# Peer review of "Parallel Microdispensing Method of High-Viscous Liquid Based on Electrostatic Force"

_micromachines, 2022, doi:10.3390/mi13040545_

Round 1

Reviewer 1 Report

Overall a very good manuscript. I recommend accept as is.

Author Response

Dear Referee

We appreciate your review of the paper. Thank you.

Reviewer 2 Report

This manuscript describes the optimization of a “smart” printing head consisting of probes that are loaded on probes using electrostatic forces, and subsequently transferred to a substrate by capillary transfer. COMSOL simulations are reported to understand and optimize the loading process, and experiments are reported of the whole printing process.

The principle of this printing method has been shown in a previous publication by the same authors, so the novelty is in its optimization (in particular by introducing serial rather than parallel loading). This is interesting for Micromachines readers, but I still recommend some revisions to include essential aspects before the manuscript can be accepted in my view.

First, the simulation focuses only on the loading. This is understandable, but the end result is also determined by the transfer process. Also, the coupling made between the simulation and the experimental results is rather weak and may be made more quantitative. I suggest:

  • To explore whether the experimental results of the loading can be semi-quantitatively compared with the simulation results.
  • To consider what determines the transfer process. For this, the liquid bridge must be broken, so I think this is determined by the interplay between surface tension of the liquid and the interfacial tension between the liquid and the substrate. Can this be made clearer or even quantitatively analyzed? Is indeed the transfer (hence the printing result) dependent on the surface energy on the substrate?

Second some small questions:

  • Table 1 indicates that the interval distance between two probes is 300 micron and the probe diameter is 300 micron. The schematic in Fig 5(a) suggestions something else, however. Can the authors explain?
  • Please add the dimensions to Fig. 5(a) – this makes the data easier to interpret.
  • Please extend the caption of Fig. 7 to explain what we see in the different panels.

Author Response

Dear Referee

Thank you for these valuable and constructive comments to greatly improve the manuscript “Parallel Micro-dispensing Method of High-Viscous Liquid based on Electrostatic Force” (ID: micromachines- 1625688). The manuscript has been carefully revised according to your suggestions.

The changes that refer to your comments are highlighted in yellow in the attached modified version. The content of manuscript is written in italics, the answers of authors are written in standard style.

Point 1: First, the simulation focuses only on the loading. This is understandable, but the end result is also determined by the transfer process. Also, the coupling made between the simulation and the experimental results is rather weak and may be made more quantitative. I suggest:

To explore whether the experimental results of the loading can be semi-quantitatively compared with the simulation results.

Response 1: According to the suggestion, the simulation results for loading process have been added to the article. Simulation of loading droplet process with different probe diameters has been added in P6 L 192~201 of Section 4.1 “The simulation of loading droplet is divided into two stages with COMSOL. The first stage is to use the electrostatic module and the phase-field module to simulate the rise of the liquid into contact with the probe. The second stage is to use the moving mesh module and the phase-field module to simulate the liquid bridge breakup process when the probe is raised. The free interface of air-liquid is defined by the isoline with 0.5 volume fraction of the viscous liquid, and the loaded volume is quantified by integral[24]. As the diameter d of the probe is 0.40 mm, 0.65 mm, 0.80 mm, and 1.00 mm, the loaded volume increases from 7.84 to 117.6 nL in the simulation. The average loaded volume increases from 7.23 to 94.94 nL, and the standard deviation is from 0.22 to 1.82 nL in the experiment. Simulation of loading droplet process with different driving voltage has been added in P6 L 205~207 of Section 4.1 “When the driving voltage increases from 400 to 650 V, the loaded volume is from 34.0 to 32.6 nL in the simulation and from 33.9 to 32.1 nL with a standard deviation of 0.22 to 0.93 nL in the experiment. Simulation of loading droplet process with different interval distance has been added in P6 L 209~213 of Section 4.1 “When the interval distance increases from 2.5 to 6.0 mm, the loaded volume slightly varies in the range from 33.8 to 32.0 nL in the simulation, and the loaded volume varies in the range from 32.30 to 31.64 nL with a standard deviation of 0.73 to 0.77 nL in the experiment, both indicating that the droplet loaded volume is less dependent on the driving voltage and interval distance.”

Point 2: To consider what determines the transfer process. For this, the liquid bridge must be broken, so I think this is determined by the interplay between surface tension of the liquid and the interfacial tension between the liquid and the substrate. Can this be made clearer or even quantitatively analyzed? Is indeed the transfer (hence the printing result) dependent on the surface energy on the substrate?

Response 2: We have previously studied the transfer process, which has been published in the article "Squeezing Dynamic Mechanism of High-Viscosity Droplet and its Application for Adhesive Dispensing in Sub-Nanoliter Resolution" (Micromachines, 2019, 10). In this article, an analytical model had been developed to describe the dynamic mechanism of squeezing and deforming a viscous droplet between plates in a transfer printing process. During the transfer process, the contact force is mainly composed of surface tension and viscous force. The viscous force of high-viscous liquid is more dominant than surface tension for nL level liquid transfer. The citation of the article was added in the P8 L 238~240 of Section 4.2“During the transfer process, the contact force is mainly composed of surface tension and viscous force. The viscous force of high-viscous liquid is more dominant than surface tension for nL level liquid transfer[23]. This paper focuses on the improvement of the device for measuring the contact force. Compared to the original device, the force sensor is installed above the probe, and the extra force balance module is developed accordingly. Thus, the smart printing head can complete the task of loading and transferring droplets, which greatly increases the integration and flexibility of the device.

Point 3: Table 1 indicates that the interval distance between two probes is 300 micron and the probe diameter is 300 micron. The schematic in Fig 5(a) suggestions something else, however. Can the authors explain?

Please add the dimensions to Fig. 5(a) – this makes the data easier to interpret.

Please extend the caption of Fig. 7 to explain what we see in the different panels.

Response 3: Sorry for the mistake. The interval distance between two probes is 3.0 mm, which has been corrected in Table 1. The dimensions have been added to Fig. 5(a).

The caption of Fig. 7 has been added in the P6 L 182~184 of Section 4.1 Figure 7. (a) The probes are moved onto the viscous liquid film; (b)(c) The liquid below the probe is stretched up by electrostatic force until it contacts the probe; (d)(e) The connection of electric power is switched to another probe; (f) The printing head is lifted to break up these liquid bridges.”

In addition, we have also polished other parts of the article, which have been highlighted in yellow in the attached modified version.

Reviewer 3 Report

In the work, the authors present an interesting and original design solution of the print head. The description of the solution is concise and precise. The authors also present the results of operational tests of the head and their analysis. The work ends with a set of logical and coherent conclusions.

In my opinion, the work was prepared correctly and carefully. It's only drawback seems to be the size of the drawings. Their size makes it very difficult to analyze their course.

My personal suggestion is to correct their size. I recommend the work for publication.

Author Response

Dear Referee

We would like to thank you for your valuable and constructive comments to greatly improve the manuscript “Parallel Micro-dispensing Method of High-Viscous Liquid based on Electrostatic Force” (ID: micromachines- 1625688). The manuscript has been carefully revised according to your suggestions.

The content of manuscript is written in italics, the answers of authors are written in standard style.

Point 1: In my opinion, the work was prepared correctly and carefully. It's only drawback seems to be the size of the drawings. Their size makes it very difficult to analyze their course.

Response 1: We have resized the drawings based on your suggestion. The size of Figure 5 is increased from 7.15×9.67 cm to 7.68×10.26 cm, and the size of Figure 8 is increased from 7.15×14.51 cm to 7.79×14.85 cm.
